# Sialyllactose Attenuates Inflammation and Injury of Intestinal Epithelial Cells upon Enterotoxigenic *Escherichia coli* Infection

**DOI:** 10.3390/ijms26083860

**Published:** 2025-04-18

**Authors:** Qiming Duan, Bing Yu, Zhiqing Huang, Yuheng Luo, Ping Zheng, Xiangbing Mao, Jie Yu, Junqiu Luo, Hui Yan, Jun He

**Affiliations:** 1Institute of Animal Nutrition, Sichuan Agricultural University, Chengdu 611130, China; qimingduan0910@163.com (Q.D.);; 2Key Laboratory for Animal Disease-Resistance Nutrition, Chengdu 611130, China

**Keywords:** sialyllactose, *Escherichia coli*, lipopolysaccharide, inflammatory injury, IPEC-J2 cells, NF-κB

## Abstract

Sialyllactose (SL), a bioactive trisaccharide abundant in porcine colostrum, demonstrates multifunctional properties including antimicrobial activity, immune regulation, and apoptosis inhibition. This research uncovers the mechanisms by which SL mitigates enterotoxigenic *Escherichia coli* (ETEC)-mediated damage to intestinal barrier integrity, employing IPEC-J2 porcine epithelial models. SL pre-treatment effectively blocked pathogen adhesion by competitively binding to cellular receptors, concurrently mitigating inflammation through significant suppression of *TNF-α*, *IL-1β*, and *IL-6* expression (*p* < 0.05). Notably, SL exhibited functional parallels to the NF-κB inhibitor BAY11-7082, jointly enhancing tight junction integrity via ZO-1 protein stabilization and inhibiting pro-inflammatory signaling through coordinated suppression of IκB-α/NF-κB phosphorylation cascades. The dual-action mechanism combines molecular interception of microbial attachment with intracellular modulation of the TLR4/MyD88/NF-κB pathway, effectively resolving both pathogenic colonization and inflammatory amplification. These findings position SL as a potential therapeutic application nutraceutical for livestock, with the capacity to address post-weaning porcine enteritis through functional feed formulations that synergistically enhance intestinal barrier resilience while curbing ETEC-mediated inflammatory pathogenesis.

## 1. Introduction

The intestinal epithelium’s defensive architecture, constituting a pivotal immunological interface, orchestrates a dual biosystemic role: facilitating bidirectional material trafficking through selective permeability for nutrient assimilation, and maintaining evolutionary conserved pathogen exclusion mechanisms against luminal xenobiotic infiltration [1]. Sustaining the architectural integrity and physiological competence of the gastrointestinal mucosal interface constitutes a cornerstone in immune-endocrine crosstalk regulation. ETEC is one of the most common external antigens [2]. It is capable of expressing one or more membrane adhesins that facilitate its attachment to the intestinal mucosa surface and strengthen its high-affinity interaction with glycoprotein receptors localized on the microvillar surface of intestinal epithelial cells. Following pathogenic colonization, ETEC secretes thermolabile (LT) and thermostable (ST) enterotoxins that activate Toll-like receptor-mediated signaling cascades. This immunopathological cascade induces aberrant overexpression of interleukin-driven inflammatory mediators and initiates caspase-dependent programmed cell death in gut epithelial lineages. The cumulative pathophysiological manifestations include compromised mucosal integrity and electrolyte transport dysregulation, clinically presenting as secretory diarrheal syndromes [3,4]. Within commercial pig production systems, neonatal scours associated with ETEC colonization pose a major health challenge during the weaning transition, which can lead to a worsened productive performance and even death in weaning piglets, resulting in significant economic losses [5]. Weaning deprives piglets of the passive immunity provided by sows’ milk, making the piglets more vulnerable to infections from enteric pathogens [6]. Thus, nutritional strategies incorporating breast milk-derived bioactive constituents into feed formulations demonstrate potential efficacy in mitigating ETEC-mediated intestinal barrier compromise.

Breast milk is rich in a variety of structurally distinct oligosaccharides, which are indigestible carbohydrates for humans and mammals, known as milk oligosaccharides (MOs) [7]. Milk oligosaccharides (MOs) consist of five fundamental monosaccharide units: galactose, glucose, fucose, N-acetylglucosamine, and the sialylated derivative N-acetylneuraminic acid. Sialyllactose (SL), the predominant structural variant within MOs, constitutes 70% of the total oligosaccharides in porcine/bovine mammary secretions and 30% in human lactation products [8]. SL is a macromolecule with a molecular weight of 633.55, consisting of an N-acetyl-D-neuraminic acid unit linked to the galactose unit of lactose through α 2,6- and α 2,3-glycosidic bonds (Appendix A) [9,10,11]. Studies have indicated that D-galactose is a central component of the ETEC receptors on the surface of intestinal epithelial cells, playing a significant role in the adhesion of ETEC strains such as K88, K99, 987P, and H10407. D-galactose exhibits the strongest affinity for ETEC K88, and compounds with similar structures, such as oligogalactose and galactosamine, can also inhibit the adhesion of ETEC to intestinal epithelial cells [12,13]. In addition, sialic acid has been proven to be involved in the adhesion processes of various bacteria and viruses [14]. For example, the adhesion of ETEC to piglet intestinal epithelial cells is highly correlated with the degree of sialylation of their brush border glycoproteins [15]. Glycoproteins modified with sialic acid can alleviate ETTC-induced inflammatory responses by inhibiting its adhesion [16]. Sialyllactose, the most abundant oligosaccharide in sow milk, contains both galactose and salic acid residues in its structure, but it is not clear whether it can similarly inhibit ETEC adherence and thereby alleviate intestinal inflammatory injury in piglets.

This investigation employed ETEC K88/LPS-challenged IPEC-J2 monolayers to delineate SL’s modulatory effects on inflammatory signaling cascades. The experimental data robustly validate SL’s capacity to suppress pro-inflammatory mediators while mechanistically elucidating its therapeutic bioactivity through TLR4/MyD88/NF-κB axis regulation.

## 2. Results

### 2.1. SL Treatment Dose and Duration

Appendix A demonstrates the dose-dependent responses of IPEC-J2 cells to SL (0–20 mg/mL), indicating the viability and transcriptional modulation of inflammatory mediators/tight junction biomarkers. A cytocompatibility assessment confirmed non-cytotoxic effects at 80 mg/mL SL (24 h exposure). A 5 mg/mL SL regimen (12 h) exhibited a dual regulatory capacity involving the suppression of IL-1β transcription and the enhancement of Occludin/ZO-1 mRNA synthesis (Appendix A, *p* < 0.05). This optimized preconditioning protocol (5 mg/mL, 12 h) was consequently implemented to counteract epithelial inflammatory pathogenesis.

### 2.2. Protective Effect of SL on Intestinal Epithelial Cells (IECs) Challenged by ETEC

As shown in Figure 1, the early-stage and total apoptosis levels in ETEC-exposed cells were markedly attenuated by SL pre-incubation (*p* < 0.05). Additionally, the qPCR quantification further revealed that Cysteine-dependent aspartate-specific protease 3, 8, and 9 (*Caspase 3*, *8*, and *9*) transcripts were markedly downregulated in ETEC-infected IPEC-J2 cells following SL preconditioning (*p* < 0.05, Figure 1B).

### 2.3. SL Alleviates Intestinal Inflammation by Preventing the Adhesion of ETEC K88 on IECs

SL preconditioning suppressed ETEC K88 adhesion to IPEC-J2 monolayers and mitigated acute inflammatory damage induced by ETEC infection, as evidenced by downregulated pro-inflammatory mediators including the following: *Tumor Necrosis Factor Alpha* (*TNF-α*), *Interleukin-1* (*IL-1β*), and *Interleukin-6* (*IL-6*), as well as up-regulation of the expression of tight-junction protein ZO-1 (*p* < 0.05, Figure 2). Prophylactic SL administration markedly downregulated transcript levels of key innate immune regulators—MyD88 (*myeloid differentiation primary response 88*), *TLR4* (*toll-like receptor 4*), and *NF-κB* (*nuclear factor-kappa B*)—in ETEC-infected IPEC-J2 cells (Figure 2B, *p* < 0.05). This transcriptional suppression of the TLR4/MyD88/NF-κB axis implies that SL-mediated anti-inflammatory effects are orchestrated through dampening canonical pathogen recognition signaling, thereby attenuating cytokine storm cascades during enterotoxigenic challenge.

### 2.4. SL Attenuated Inflammation and Injury of the IECs Challenged by LPS

SL pre-treatment improved the integrity of the LPS-challenged cells, as indicated by the increased distribution and abundance of the tight-junction protein ZO-1 (*p* < 0.05, Figure 3A). Concomitantly, SL preconditioning attenuated pro-inflammatory mediator release, demonstrating significant reductions in *TNF-α*, *IL-1β*, and *IL-6* compared to the ETEC-challenged group (*p* < 0.05, Figure 3B).

### 2.5. Protective Effect of SL on IECs Challenged by LPS

The results of flow cytometry analysis revealed that SL pre-treatment significantly decreased the percentages of early-stage, late-stage, and total apoptotic cells upon LPS challenge (*p* < 0.05, Figure 4A). Meanwhile, SL pre-treatment significantly decreased the mRNA expression levels of *Caspase 3* and *9* in LPS-treated IPEC-J2 cells (*p* < 0.05, Figure 4B).

### 2.6. SL Inhibits the Phosphorylation of IκB-α and NF-κB by Preventing the Adhesion of LPS on IECs

LPS mono-stimulation induced robust membrane-localized fluorescence indicative of pathogen-associated molecular pattern binding on IPEC-J2 monolayers, whereas SL coadministration markedly attenuated surface ligand clustering intensity (*p* < 0.05, Figure 5A). Immunoblotting demonstrated SL-mediated suppression of canonical inflammatory signaling, with dose-dependent attenuation of LPS-triggered phosphorylation events in both IκBα and NF-κB p65 (*p* < 0.05, Figure 5B).

### 2.7. SL and BAY 11-7082 Attenuated Inflammation and Injury of the IECs Challenged by LPS

SL-mediated preservation of the tight junction architecture was correlated with ZO-1 transcriptional activation, achieving comparable efficacy to the canonical NF-κB inhibitor BAY 11-7082 in restoring epithelial barrier function (*p* < 0.05, Figure 6A). Meanwhile, SL and BAY 11-7082 pre-treatment significantly increased the mean optical density of ZO-1 protein LPS-treated IPEC-J2 cells (*p* < 0.05, Figure 6A). Furthermore, LPS challenge induced robust transcriptional activation of canonical pro-inflammatory mediators (*TNF-α*, *IL-1β*, and *IL-6*) in porcine enterocytes (*p* < 0.05, Figure 6B). In contrast, pre-treatment with SL or BAY 11-7082 markedly reduced the mRNA expression of these pro-inflammatory cytokines (*p* < 0.05, Figure 6B).

### 2.8. SL and BAY 11-7082 Inhibit Apoptosis on IECs Challenged by LPS

Flow cytometry results revealed that SL and BAY 11-7082 pre-treatment significantly decreased the percentages of early-stage and total apoptotic cells in LPS-treated IPEC-J2 cells (*p* < 0.05, Figure 7A). Concurrently, LPS treatment significantly increased the transcript abundance of *Caspase 3*, *Caspase 8*, and *Caspase 9*, while SL and BAY 11-7082 pre-treatment significantly inhibited *Caspase 3*, *Caspase 8*, and *Caspase 9* mRNA expression (*p* < 0.05, Figure 7B).

### 2.9. SL and BAY 11-7082 Decreased the Phosphorylation of the Critical Inflammation-Associated Proteins IκB-α and NF-κB in LPS-Challenged Cells

Western blot analysis revealed that SL exerts a similar effect to BAY 11-7082, and both can decrease the abundance of phosphorylation of the NF-κB and IκBα proteins in LPS-treated cells (*p* < 0.05, Figure 8). These results indicate that SL-mediated suppression of canonical NF-κB nuclear translocation in LPS-stimulated porcine intestinal epithelium can achieve comparable IκBα stabilization efficacy to the pharmacological comparator BAY 11-7082.

## 3. Discussion

Within porcine production systems, ETEC-associated infections incur substantial economic losses, predominantly through mortality escalation and growth performance reduction. Post-weaning piglets subjected to ETEC colonization typically demonstrate elevated apoptotic indices in IECs [17]. The current experimental data corroborate this pathogenic mechanism, revealing significantly elevated apoptotic indices in ETEC-colonized IPEC-J2 monolayers versus unchallenged controls. Prophylactic SL administration demonstrated substantial anti-apoptotic efficacy in pathogen-exposed models. Molecular profiling further confirmed SL-mediated transcriptional suppression of caspase cascade effectors (*caspase-3*/*-8*/*-9*) in enteropathogen-stimulated epithelial cells. This collective evidence establishes SL’s cytoprotective functionality against ETEC-induced IEC pathophysiology.

ETEC K88 fimbriae exhibit stereoselective binding to sialoglycoconjugates, intestinal mucin-type glycoproteins, and neutral glycosphingolipids via α/β-galactosidic linkages. This microbial adhesion can be competitively inhibited by oligosaccharide structural analogs mimicking host glycan motifs. Such molecular interference effectively blocks microbial adhesion to epithelial membranes, consequently attenuating pathogen virulence potential [18,19]. Sialyllactose contains both galactose and sialic acid moiety (Neu5Ac) [20]. Studies have shown that SL has an antiadhesive antimicrobial ability, which can inhibit the adhesion of *Clostridioides difficile*, *S. enterica* subsp, and *E. coli* O119 to intestinal epithelial cells [21,22]. Consistent with these results, prophylactic SL administration attenuated enteropathogen colonization efficiency, demonstrating a reduction in ETEC K88 fimbrial attachment density on polarized porcine enterocytes compared to untreated challenge controls. TLR4-mediated MyD88-dependent signaling constitutes an essential pathogenic interface wherein ETEC subverts host immunosurveillance, propelling NF-κB-driven pro-inflammatory cascade via FliC flagellin recognition. Functioning as an innate immune sensor, TLR4 identifies microbial signature molecules on ETEC surfaces, initiating the transduction of effector signaling cascades including the MyD88-mediated signaling axis [23]. Ligand engagement induces TLR4-MyD88 heterodimeric assembly, enabling sequential kinase mobilization through IRAK effector recruitment within the interleukin-1 receptor signaling cascade [24]. TLR4 activation initiates kinase signaling cascades, culminating in NF-κB phosphorylation. Nuclear translocation of activated NF-κB induces transcriptional activation of inflammatory mediators—encompassing cytokine/chemokine biosynthesis pathways and cell adhesion machinery—integral to the host’s defense mechanisms against ETEC pathogenesis [23]. In the present study, SL administration mitigated ETEC-induced inflammatory pathogenesis in intestinal epithelia, as evidenced through the transcriptomic suppression of TNF-α/IL-1β/IL-6 cytokine clusters. Concomitant down-modulation of TLR4 receptor expression, MyD88 adaptor protein expression, and NF-κB transcriptional activity mechanistically delineates SL’s barrier-protective effects via blockade of the TLR4-mediated MyD88/NF-κB signaling axis.

Disequilibrium in cytokine homeostasis (anti-/pro-inflammatory) precipitates pathogenic disruption of the integrity of the intestinal barrier—a structure whose molecular foundation resides in intercellular junction complexes comprising occludin, claudin isoforms, and zonula occludens proteins within epithelial monolayers [17,25]. Within junctions with complex architecture, ZO-1 functions as a scaffolding orchestrator, dynamically coordinating epithelial programmed cell death, cellular expansion, and ultrastructural polarization through transmembrane junctional organization [26]. In the present study, immunofluorescence quantification demonstrated that SL preconditioning attenuated ETEC-induced ZO-1 delocalization. These results clearly demonstrate that SL exerts a protective and anti-inflammatory influence on intestinal epithelial cells when challenged with ETEC.

LPS, a structural hallmark of gram-negative bacterial membranes, serves as a standard immunological trigger to activate pro-inflammatory signaling pathways across differentiated epithelial models, notably porcine IPEC-J2 enterocyte lineages [27,28,29]. A recent study revealed that the overproduction of pro-inflammatory cytokines is consistently linked to the disruption of tight junction proteins and the worsening of apoptosis in IPEC-J2 cells [30]. In a consistent manner, SL pre-treatment was found to markedly lower the expression of the pro-inflammatory cytokines TNF-α, IL-1β, and IL-6. Moreover, it enhanced the integrity of LPS-challenged cells, as indicated by an increase in ZO-1 protein abundance, and concurrently decreased the percentages of early-stage and total apoptotic cells, along with the mRNA expression levels of *Caspase 3* and *9*. Experimental evidence delineates SL’s cytoprotective mechanism in porcine enterocytes, where LPS-triggered inflammatory pathogenesis is ameliorated via dual modulatory effects: (1) the fortification of intercellular junction integrity and (2) the inhibition of programmed cell death through suppressed biosynthesis of IL-6/TNF-α/IL-1β inflammatory mediators.

LPS can bind to the TLR4 receptor on the cell surface and activate the downstream NF-κB-mediated inflammatory cascade [28,31]. Studies have shown that SL can bind directly to the LPS-binding site of the TLR4-MD2 complex, effectively preventing the binding of LPS to TLR4 and inhibiting TLR4-mediated signal transduction [32]. Consistently, immunofluorescence analysis results have revealed that SL coadministration significantly reduces the intensity of surface ligand clustering, thereby effectively inhibiting the adhesion of LPS to the surface of IPEC-J2 cells. Meanwhile, SL pre-exposure has attenuated phospho-activation of NF-κB p65 and IκBα degradation in inflamed epithelia, establishing the molecular basis for SL’s anti-inflammatory efficacy through interference with pathoadhesive interactions.

To delineate SL’s anti-inflammatory properties, we employed a pharmacological intervention with BAY11-7082—a selective suppressor of IκB-α degradation and NF-κB nuclear translocation—to determine its modulatory effects on NF-κB-dependent inflammatory cascades in intestinal epithelia. NF-κB functions as a heterodimeric transcription regulator governing the immunogenetic networks encompassing leukocyte recruitment, apoptosis modulation, and oncogenic transformation through bidirectional control of chemokine/effector protein expression [33,34]. Under basal conditions, NF-κB heterodimers exhibit cytoplasmic quiescence through constitutive binding to IκB regulatory proteins, with IκBα being the predominant isoform. This molecular sequestration effectively precludes nucleocytoplasmic trafficking, thereby maintaining transcriptional dormancy by blocking access to chromatin remodeling complexes [35]. Upon cellular stimulation, IκB-α is phosphorylated and rapidly degraded, which results in the activation of NF-κB. This activation facilitates the translocation of NF-κB into the nucleus, where it triggers the transcription and expression of genes associated with inflammation [36,37]. In addition, SL also exerts a similar effect to BAY (a specific inhibitor of NF-κB) and can both improve the distribution and abundance of ZO-1 in LPS-challenged cells, and suppress apoptosis and the production of inflammatory cytokines in the cells. Additionally, both SL and BAY have notably reduced the phosphorylation of key inflammation-related proteins, including IκB-α and NF-κB, in LPS-stimulated cells. These results provide further evidence supporting the hypothesis that SL could function as an anti-inflammatory agent.

## 4. Materials and Methods

### 4.1. Bacterial Strains and Culture

The source and culture of pathogenic *Escherichia coli* (ETEC) were referenced from previous studies conducted in our laboratory [38].

### 4.2. Sialylactose

Sialylactose (SL, ≥95%) with a formula of C_23_H_39_NO_19_ and a molecular weight of 633.55 was donated by Glycom A/S and its structure is shown in Appendix A. SL is a trisaccharide produced through microbial fermentation, composed of glucose, galactose, and N-acetylneuraminic acid.

### 4.3. Cell Culture

The cell culture methods and reagents used were referenced from the study by Wan et al. [17]. The cells were subcultured every 2–3 days at a 1:3 ratio when they reached around 80–90% confluence.

### 4.4. ETEC Treatment

The experimental protocols and reagents used during ETEC intervention were adapted from the methodology established in Fu et al.’s prior research [31]. The study comprised four experimental conditions:

(1) Baseline control (CON) with neither SL pre-treatment nor ETEC exposure;

(2) SL-only treatment (CSL) involving 12 h pre-incubation with 5 mg/mL SL;

(3) ETEC challenge (ETEC) using 1 × 10^6^ CFU/well bacterial suspension applied for 1 or 2.5 h;

(4) Combined intervention (ESL) combining 5 mg/mL SL pre-treatment followed by dual-phase ETEC exposure.

Following these interventions, both cellular material and conditioned media were collected for subsequent analytical procedures.

### 4.5. LPS Treatment

The dose and duration of LPS treatment were determined using previous studies carried out in our laboratory [17]. A factorial experimental design was implemented, comprising four distinct cohorts: a control (CON) cohort that received neither SL preconditioning nor LPS stimulation; a CSL cohort that was subjected to 12 h pharmacological preconditioning with 5 mg/mL SL; an LPS cohort that was exposed to 6 h endotoxin challenge (5 μg/mL); and a sequential intervention cohort (LSL) that underwent SL pre-treatment followed by LPS provocation. Post-intervention cellular monolayers and conditioned media were synchronously harvested for downstream multi-omics profiling.

### 4.6. BAY 11-7082 Treatment

The pharmacological concentrations and exposure periods for LPS and BAY 11-7082 interventions were determined according to established experimental parameters in our institutional research protocol [17]. Eight distinct experimental cohorts were designed:

(1) Negative control (CON) without any pre-treatment or stimuli;

(2) SL monotherapy (CSL) receiving 12 h preconditioning with 5 mg/mL SL;

(3) LPS stimulation (LPS) with 6 h 5 μg/mL exposure;

(4) BAY inhibitor control (BAY) undergoing 2 h 1 μmol/L pre-treatment;

(5) SL-LPS sequential administration (LSL) combining SL preconditioning and subsequent LPS challenge;

(6) BAY-LPS combined regimen (LBAY) pairing inhibitor pre-treatment with LPS stimulation.

Post-interventional specimens including cellular monolayers and corresponding conditioned media were systematically harvested for downstream analytical processing.

### 4.7. Assay of ETEC K88 Adhesion onto IPEC-J2 Cells

The quantification of ETEC adhesion on IPEC-J2 monolayers was conducted using an optimized protocol derived from Letourneau’s bacterial adherence quantification assay [39]. Post-incubation specimens underwent viable count analysis through serial decimal dilution plating on selective media. Adherent pathogen load was calculated by logarithmic conversion of mean colony counts from diagnostically valid dilution gradients (30–300 colonies/plate threshold), with parallel determination of initial inoculum concentration via control plate enumeration.

### 4.8. Immunofluorescence Assay

IPEC-J2 monolayers were established on sterile glass coverslips within 6-well plates (2 × 10^5^ cells/well initial seeding density), with culture progression monitored until 80–90% confluency was achieved. Experimental cohorts underwent differential pre-treatment protocols: 12 h incubation with 5 mg/mL SL or 2 h exposure to 1 μM BAY 11-7082 (NF-κB inhibitor), followed by 6 h LPS challenge (5 μg/mL). Subsequent fluorescent immunolocalization of tight junction protein ZO-1 and bacterial endotoxin was conducted according to the standardized immunofluorescence protocol established by Wan et al. [17].

### 4.9. RNA Extraction and Quantitative Real-Time PCR (qPCR)

Cellular RNA extraction from IPEC-J2 monolayers was performed with RNAiso Plus reagent (Takara Bio, Shiga, Japan), strictly adhering to manufacturer-specified protocols. RNA integrity verification included spectrophotometric assessment (NanoDrop 2000 system; Thermo Fisher, Waltham, MA, USA) through dual-wavelength (260/280 nm) absorbance quantification. Reverse transcription of RNA templates was executed with the PrimeScript RT Master Mix (Takara Bio) with strict adherence to the manufacturer-specified thermocycler protocol, ensuring precise cDNA strand initiation fidelity. Quantitative gene expression profiling followed the qPCR protocol established in Wan et al.’s methodology [17], with 2^−ΔΔCt^ calculations [40] applied for relative quantification against the housekeeping gene β-actin. The oligonucleotide primer sequences utilized in this study are comprehensively delineated in Table 1.

### 4.10. Detection of Cell Apoptosis

Following enzymatic dissociation of IPEC-J2 monolayers using 0.25% serum-free trypsin solution (EDTA-free formulation), cellular pellets were obtained through centrifugation (350× *g*, 10 min, 4 °C) under refrigerated conditions. Subsequent washing procedures involved dual cycles of pre-chilled phosphate-buffered saline (PBS, pH 7.4) resuspension. For apoptosis quantification, cell aliquots (100 μL suspension) were sequentially stained with 5 μL PE-conjugated Annexin V and 5 μL 7-AAD viability dye, followed by a 15 min incubation period under light-protected ambient conditions. Reaction termination was achieved by the addition of 400 μL 1× Annexin binding buffer. Fluorescence-based apoptotic profiling was executed within 60 min post-staining using a Beckman Coulter CytoFlex flow cytometry system (Beckman Coulter Life Sciences, Brea, CA, USA) and BD-FACSVerse platform (BD Biosciences, San Jose, CA, USA), with compensation controls applied during acquisition.

### 4.11. Total Protein Extraction and Western Blot Analysis

Polarized IPEC-J2 monolayers underwent protein fractionation employing ice-cold RIPA buffer (25 mM Tris-HCl pH 7.6, 150 mM NaCl, 1% NP-40, 1% sodium deoxycholate, 0.1% SDS) fortified with protease-phosphatase inhibitor cocktails, with cell debris pelleted via high-speed centrifugation (12,000× *g*, 15 min, 4 °C) to yield clarified lysate supernatants. An integrated workflow encompassing biomolecular separation, immunodetection parameters, and buffer stoichiometry was implemented in accordance with Wan et al.’s refined proteomic methodology [17]. The target-specific immunodetection reagents with corresponding dilution parameters are systematically cataloged in Table 2.

### 4.12. Statistics Analysis

Quantitative data processing was executed in IBM SPSS Statistics (v26.0, Armonk, NY, USA), with the experimental outcomes expressed as the arithmetic mean ± SEM from six biological replicates per experimental group. A two-factor factorial ANOVA model with interaction terms was implemented for hypothesis testing under normality and homogeneity of variance assumptions, followed by Fisher’s Least Significant Difference multiple comparison correction. Probabilistic thresholds adhered to the Neyman–Pearson framework, with α = 0.05 defining statistical significance. Data visualization was performed in GraphPad Prism (v8.0.2, La Jolla, CA, USA) following Wilkinson’s grammar of graphics principles for scientific illustration.

## 5. Conclusions

Collectively, these data establish SL’s therapeutic efficacy against ETEC-mediated pathogenic insults in intestinal mucosa through dual modulation of inflammatory signaling and programmed cell death cascades. Mechanistic interrogation reveals an SL-mediated blockade of LPS’s pathoadhesive interactions with the TLR4/MD2 receptor complex, culminating in the attenuated phosphorylation of NF-κB transcriptional regulators and the subsequent transcriptional quiescence of the IL-6/IL-1β/TNF-α triad biosynthesis. This pharmacological profile highlights SL as a novel nutraceutical agent for intervention in bacterial enteritis pathologies across mammalian systems, particularly in swine model systems exhibiting compromised intestinal barrier function.

## Figures and Tables

**Figure 1 ijms-26-03860-f001:**
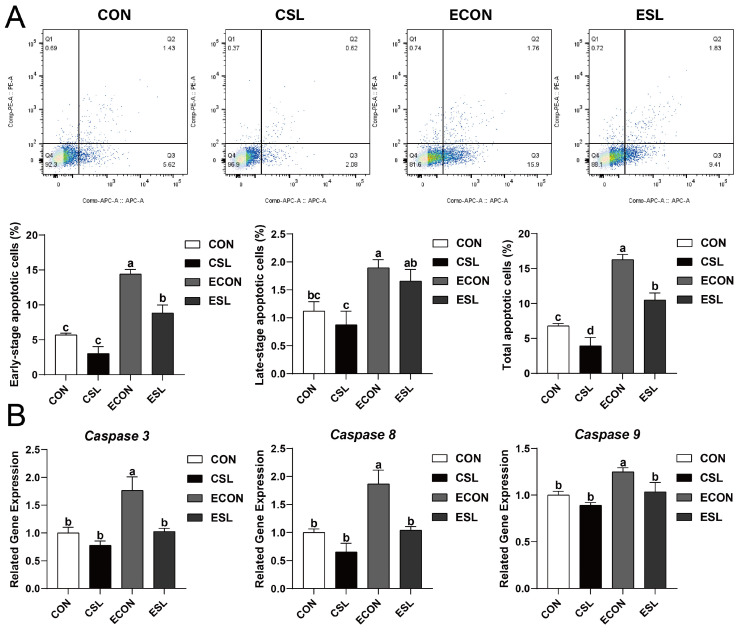
The protective effect of SL on intestinal epithelial cells (IECs) challenged by ETEC. (**A**) Apoptotic cells detected by flow cytometry. Blue = individual cells; Green = clusters of multiple cells; Orange = dense aggregates of numerous cells. (**B**) Quantitative PCR analysis using SYBR Green chemistry revealed the regulation of *Caspase 3*, *Caspase 8*, and *Caspase 9* in IPEC-J2 cells with relative expression normalized to GAPDH via 2^−ΔΔCt^ methodology. CON: control group, CSL: SL group, ECON: ETEC group, ESL: ETEC + SL group. For *Caspase 3*, *8*, and *9*, all values are expressed as the mean ± SEM (*n* = 6). Significant differences (*p* < 0.05) are denoted by superscript letters (a–d) across treatment groups.

**Figure 2 ijms-26-03860-f002:**
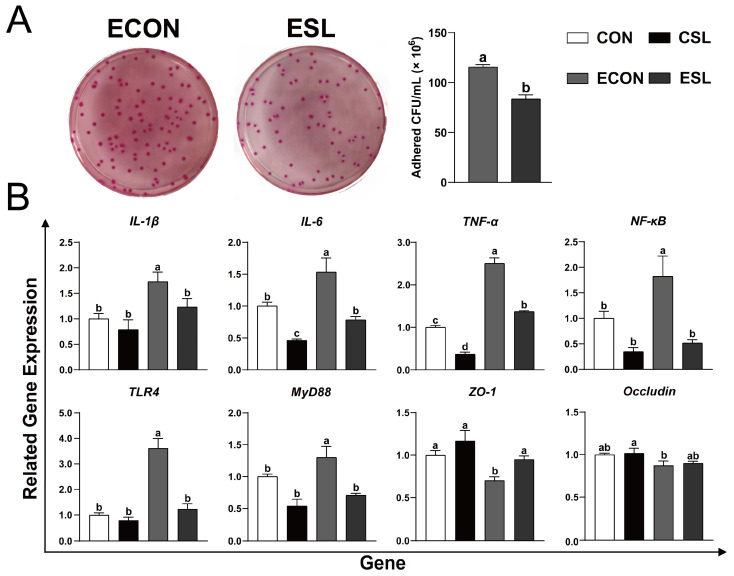
SL alleviates intestinal inflammation by preventing the adhesion of ETEC K88 on IECs. (**A**) The number of ETEC K88 cfu adhered to the IPEC-J2 cell surface. (**B**) Quantitative PCR analysis using SYBR Green chemistry revealed the regulation of *IL-1β*, *IL-6*, *MyD88*, *NF-κB*, *Occludin*, *TNF-α*, *TLR4*, and *ZO-1* in IPEC-J2 cells with relative expression normalized to GAPDH via 2^−ΔΔCt^ methodology. CON: control group, CSL: SL group, ECON: ETEC group, ESL: ETEC + SL group. All values are expressed as the mean ± SEM (*n* = 6). Significant differences (*p* < 0.05) are denoted by superscript letters (a–d) across treatment groups.

**Figure 3 ijms-26-03860-f003:**
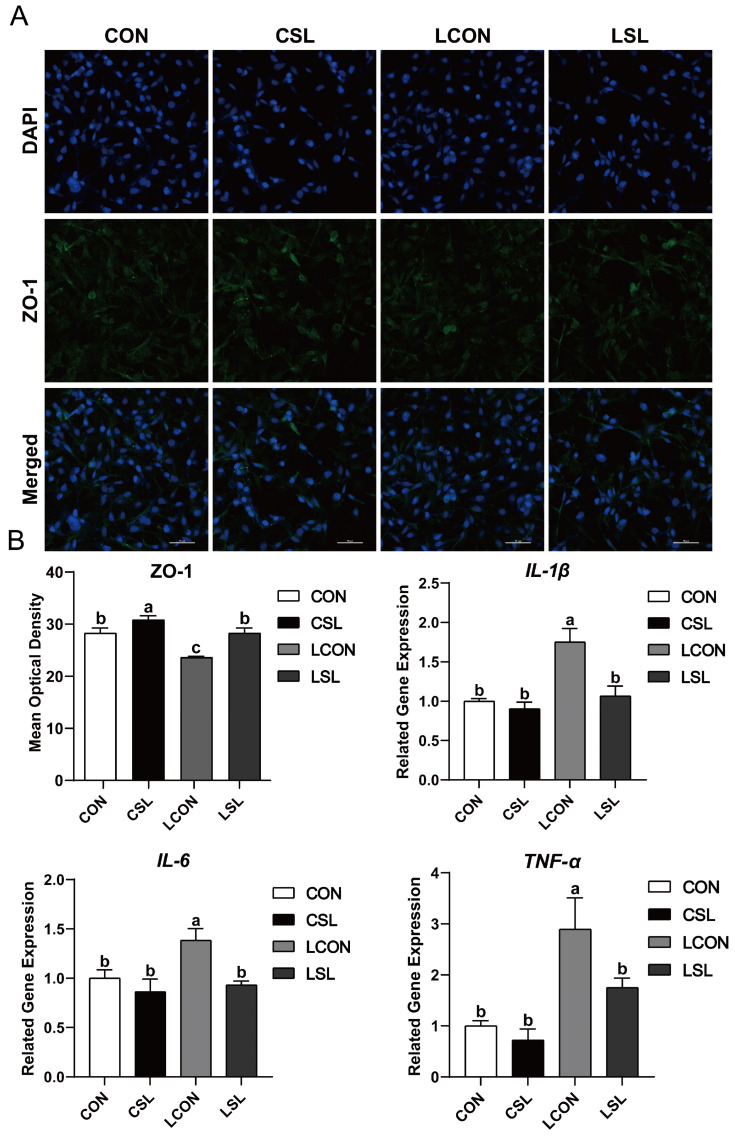
SL attenuated inflammation and injury of the IECs challenged by LPS. (**A**) Representative images of ZO-1 using an immunofluorescence assay (captured using a laser scanning confocal microscope; 400×). Scale bar: 50 μm. (**B**) Quantitative PCR analysis using SYBR Green chemistry revealed the regulation of *IL-1β*, *IL-6*, and *TNF-α* in IPEC-J2 cells with relative expression normalized to GAPDH via 2^−ΔΔCt^ methodology. CON: control group, CSL: SL group, LCON: LPS group, LSL: LPS + SL group. All values are expressed as the mean ± SEM (*n* = 6). Significant differences (*p* < 0.05) are denoted by superscript letters (a–c) across treatment groups.

**Figure 4 ijms-26-03860-f004:**
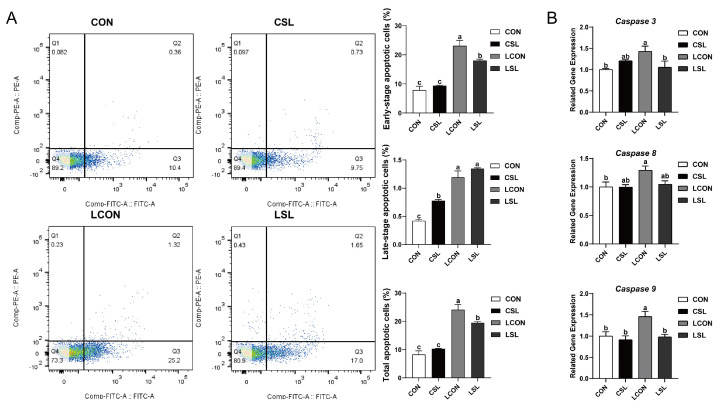
The protective effect of SL on intestinal epithelial cells (IECs) challenged by LPS. (**A**) Apoptotic cells detected by flow cytometry. Blue = individual cells; Green = clusters of multiple cells; Orange = dense aggregates of numerous cells. (**B**) Quantitative PCR analysis using SYBR Green chemistry revealed the regulation of *Caspase 3*, *Caspase 8*, and *Caspase 9* mRNA in IPEC-J2 cells with relative expression normalized to GAPDH via 2^−ΔΔCt^ methodology. CON: control group, CSL: SL group, LCON: LPS group, LSL: LPS + SL group. All values are expressed as the mean ± SEM (*n* = 6). Significant differences (*p* < 0.05) are denoted by superscript letters (a–c) across treatment groups.

**Figure 5 ijms-26-03860-f005:**
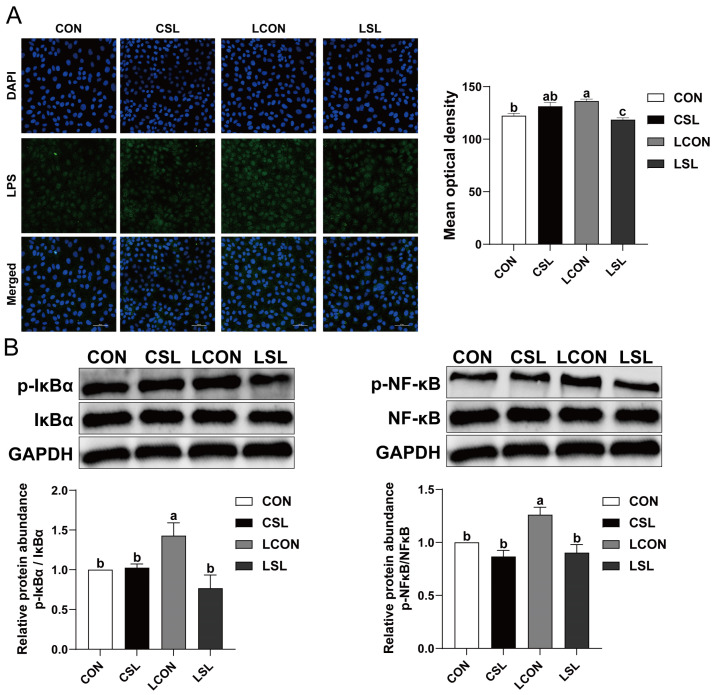
SL inhibits the phosphorylation of IκB-α and NF-κB by preventing the adhesion of LPS on IECs. (**A**) LPS ligand-receptor docking efficiency at the porcine intestinal epithelial interface. Scale bar: 50 μm. (**B**) The protein abundances of IκB-α, p-IκB-α, NF-κB, and p-NF-κB. CON: control group, CSL: SL group, LCON: LPS group, LSL: LPS + SL group. All values are expressed as the mean ± SEM (*n* = 6). Significant differences (*p* < 0.05) are denoted by superscript letters (a–c) across treatment groups.

**Figure 6 ijms-26-03860-f006:**
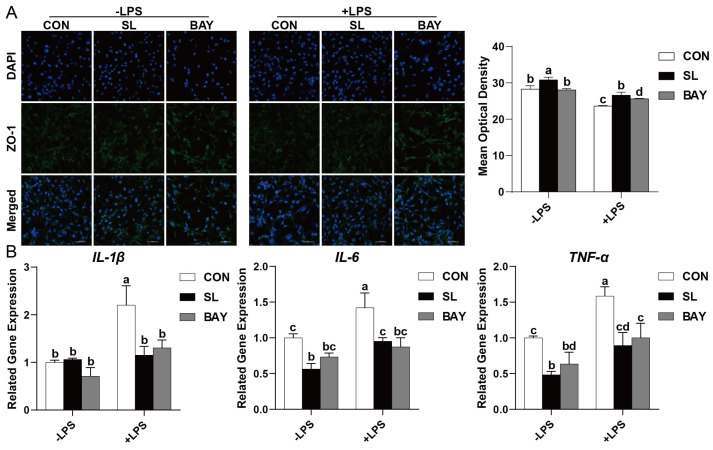
SL and BAY 11-7082 attenuate inflammation and injury of the IECs challenged by LPS. (**A**) Representative images of ZO-1 from an immunofluorescence assay (captured using a laser scanning confocal microscope; 400×). Scale bar: 50 μm. (**B**) Quantitative PCR analysis using SYBR Green chemistry revealed the regulation of *IL-1β*, *IL-6*, and *TNF-α* mRNA in IPEC-J2 cells with relative expression normalized to GAPDH via 2^−ΔΔCt^ methodology. CON: cells without SL or BAY 11-7082 pre-treatment or LPS treatment, SL: cells pre-treated with SL, BAY: cells pre-treated with BAY 11-7082, -LPS: cells without LPS treatment, +LPS: cells pre-treated with LPS., All values are expressed as the mean ± SEM (*n* = 6). Significant differences (*p* < 0.05) are denoted by superscript letters (a–d) across treatment groups.

**Figure 7 ijms-26-03860-f007:**
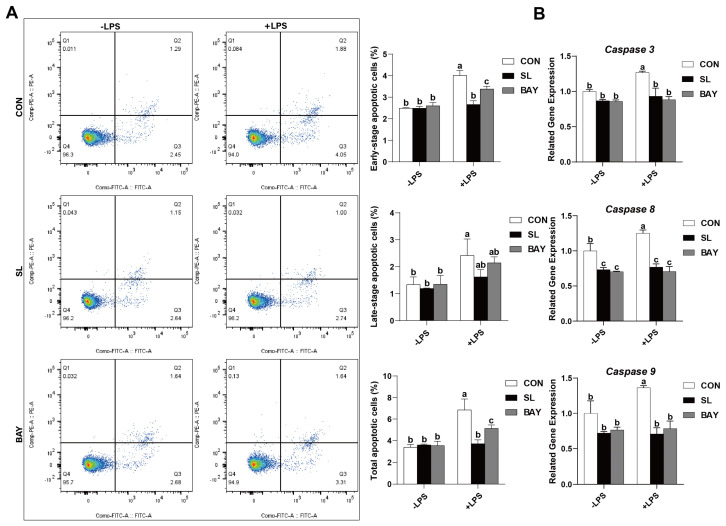
SL and BAY 11-7082 inhibit apoptosis on the IECs challenged by LPS. (**A**) Apoptotic cells detected by flow cytometry. Blue = individual cells; Green = clusters of multiple cells; Orange = dense aggregates of numerous cells. (**B**) Quantitative PCR analysis using SYBR Green chemistry revealed the regulation of *Caspase 3*, *Caspase 8*, and *Caspase 9* mRNA in IPEC-J2 cells with relative expression normalized to GAPDH via 2^−ΔΔCt^ methodology. CON: cells without SL or BAY 11-7082 pre-treatment or LPS treatment, SL: cells pre-treated with SL, BAY: cells pre-treated with BAY 11-7082, -LPS: cells without LPS treatment, +LPS: cells pre-treated with LPS. All values are expressed as the mean ± SEM (*n* = 6). Significant differences (*p* < 0.05) are denoted by superscript letters (a–c) across treatment groups.

**Figure 8 ijms-26-03860-f008:**
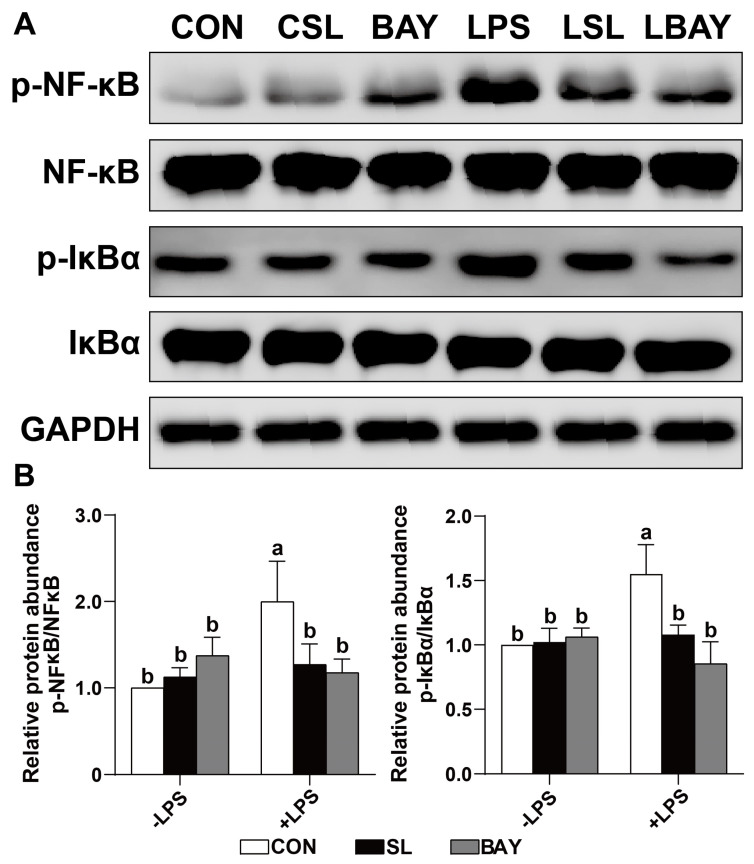
SL and BAY 11-7082 prevent the phosphorylation of the IκB-α and NF-κB of IECs challenged by LPS. (**A**) The protein abundances of IκB-α and p-IκB-α. CON: cells without SL or BAY 11-7082 pre-treatment or LPS treatment, CSL: cells pre-treated with SL, BAY: cells pre-treated with BAY 11-7082, LPS: cells pre-treated with LPS, LSL: cells pre-treated with SL and subsequently treated with LPS, LBAY: cells pre-treated with BAY 11-7082 and subsequently treated with LPS. (**B**) The protein abundances of NF-κB and p-NF-κB. CON: cells without SL or BAY 11-7082 pre-treatment or LPS treatment, SL: cells pre-treated with SL, BAY: cells pre-treated with BAY 11-7082, -LPS: cells without LPS treatment, +LPS: cells pre-treated with LPS. All values are expressed as the mean ± SEM (*n* = 6). Significant differences (*p* < 0.05) are denoted by superscript letters (a,b) across treatment groups.

**Table 1 ijms-26-03860-t001:** Sequences of primers for genes.

Gene *	Primer Sequence (5′–3′)	Product Size (bp)
*β-Actin*	F: TGGAACGGTGAAGGTGACAGC	177
	R: GCTTTTGGGAAGGCAGGGACT	
*Caspase 3*	F: GGGATTGAGACGGACAGTGG	136
	R: TGAACCAGGATCCGTCCTTTG	
*Caspase 8*	F: TCTGCGGACTGGATGTGATT	165
	R: TCTGAGGTTGCTGGTCACAC	
*Caspase 9*	F: AATGCCGATTTGGCTTACGT	195
	R: CATTTGCTTGGCAGTCAGGTT	
*IL-1β*	F: GTGATGCCAACGTGCAGTCT	97
	R: AGGTGGAGAGCCTTCAGCAT	
*IL-6*	F: TGGCTACTGCCTTCCCTACC	153
	R: CACACATCTCCTTTCTCATTGC	
*MyD88*	F: CCATTCGAGATGACCCCCTG	183
	R: TAGCAATGGACCAGACGCAG	
*NF-κB*	F: GTGTGTAAAGAAGCGGGACCT	139
	R: CACTGTCACCTGGAAGCAGAG	
*Occludin*	F: CTACTCGTCCAACGGGAAAG	158
	R: ACGCCTCCAAGTTACCACTG	
*TNF-α*	F: GCATCGCCGTCTCCTACCAG	173
	R: GGGCAGGTTGATCTCGGCAC	
*TLR4*	F: TTACAGAAGCTGGTTGCCGT	152
	R: TCCAGGTTGGGCAGGTTAGA	
*ZO-1*	F: CAGCCCCCGTACATGGAGA	114
	R: GCGCAGACGGTGTTCATAGTT	

* IL-1β: Interleukin-1 beta, IL-6: Interleukin-6, MyD88: Myeloid Differentiation Primary Response Protein 88, NF-κB: nuclear factor kappa B, TLR4: Toll-like receptor 4, TNF-α: Tumor Necrosis Factor Alpha, ZO-1: Zonula Occludens-1 Protein.

**Table 2 ijms-26-03860-t002:** Antibody information.

Name	Supplier and Catalog Number	Dilution Factor
Rabbit anti-ZO-1	Abcam plc. (Cambridge, UK)	1:150
Mouse anti-*E. coli* LPS	Abcam plc. (Cambridge, UK)	1:50
FITC-conjugated goat anti-rabbit IgG antibody	Abcam plc. (Cambridge, UK)	1:2500
TRITC-conjugated goat anti-mouse IgG	Abcam plc. (Cambridge, UK)	1:2500
p-NF-κB p65	Cell Signaling Technology 3033S (Danvers, MA, USA)	1:1000
NF-κB p65	Cell Signaling Technology 6956S (Danvers, MA, USA)	1:1000
p-IκBα	Invitrogen MA5-15224 (Danvers, MA, USA)	1:1000
IκBα	Cell Signaling Technology4814S (Danvers, MA, USA)	1:1000
GAPDH	Cell Signaling Technology 2118S (Danvers, MA, USA)	1:1000
Anti-rabbit IgG	Cell Signaling Technology 7074S (Danvers, MA, USA)	1:2500
Anti-mouse IgG	Cell Signaling Technology 7076S (Danvers, MA, USA)	1:2500

## Data Availability

The data are all included in the manuscript.

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
