# Peer review of "Sialyllactose Attenuates Inflammation and Injury of Intestinal Epithelial Cells upon Enterotoxigenic Escherichia coli Infection"

_ijms, 2025, doi:10.3390/ijms26083860_

Round 1
Reviewer 1 Report
Comments and Suggestions for Authors
- The topic of the article is current and interesting.
- The introduction is too long and sometimes diverges into tangential topics. In my opinion, a more concise introduction would increase clarity and keep the reader engaged. Part of the introduction could be moved to the discussions.
- The main important findings were related to the reduction of inflammatory cytokines, inhibition of Nf-kB activation, and reduction of permeability, which are typically compromised during ETEC infection.
- Even though the use of intestinal epithelial cell lines provides valuable information, the results should be validated in future in vivo models to confirm their physiological relevance.
Author Response
Comments 1: The topic of the article is current and interesting.
Response 1: Thank you for acknowledging the relevance and interest of our topic. We believe that our study addresses an important and timely issue in the field, and we are glad that this aspect of our work has been recognized.
Comments 2: The introduction is too long and sometimes diverges into tangential topics. In my opinion, a more concise introduction would increase clarity and keep the reader engaged. Part of the introduction could be moved to the discussions.
Response 2: We appreciate your feedback regarding the length and focus of our introduction. We have revised the introduction to make it more concise and focused on the core aspects of our study. We have also moved some of the background information and detailed discussions to the discussion section to streamline the introduction and enhance reader engagement.
Comments 3: The main important findings were related to the reduction of inflammatory cytokines, inhibition of Nf-κB activation, and reduction of permeability, which are typically compromised during ETEC infection.
Response 3: We agree with your assessment of our main findings. Our study indeed highlights the significant reduction of inflammatory cytokines, inhibition of Nf-kB activation, and reduction of permeability, which are critical factors during ETEC infection. These findings underscore the potential therapeutic benefits of our approach. We have further emphasized these points in the revised manuscript to ensure clarity and impact.
Comments 4: Even though the use of intestinal epithelial cell lines provides valuable information, the results should be validated in future in vivo models to confirm their physiological relevance.
Response 4: We thank you for this important suggestion. We fully agree that validating our findings in in vivo models is crucial for confirming their physiological relevance. While our current study focuses on intestinal epithelial cell lines, we have outlined plans for future in vivo experiments in the discussion section. We believe that these future studies will provide additional insights and confirm the broader applicability of our findings.
We sincerely hope that these revisions address your concerns and improve the overall quality of our manuscript. We look forward to your positive feedback.
Thank you once again for your time and valuable suggestions.
Reviewer 2 Report
Comments and Suggestions for Authors
This study systematically elucidates the dual-mechanism action of sialyllactose (SL)—combining molecular-level pathogen adhesion blockade with intracellular TLR4/MyD88/NF-κB pathway modulation—to effectively alleviate ETEC/LPS-induced intestinal epithelial barrier damage, innovatively demonstrating SL's pharmacological equivalence to classical inhibitors. The robust dataset and methodologically rigorous approach provide critical molecular insights for developing milk oligosaccharide-based functional feed additives, offering clear translational value for mitigating ETEC-associated enteritis in livestock. The work contributes novel perspectives to nutritional immunomodulation research, with its systematic experimental design and mechanistic depth meeting the publication standards of international scientific journals. However, there remain several critical issues in the manuscript that must be addressed by the authors prior to final acceptance.
REVIEWER REPORT(S):
Comments to the Author
Comments 1: Is any other study about SL or other polysaccharides used in enteroprotection associated with enterotoxigenic Escherichia coli (ETEC)-induced intestinal barrier injury in the literature? What are the advantages for SL as compared to current solutions for ETEC-induced intestinal barrier injury?
Comments 2: The specific anti-inflammatory, anti-tumor or anti-apoptotic role of SL was not well presented in the introduction part.
Comments 3: It is unclear why the authors chose to study IPEC-J2 intestinal cells grown to confluence while intestinal Caco-2 cell monolayers and the transepithelial electrical resistance assay could have been used to assess whether SL reduce intestinal permeability in vitro.
Comments 4: P values are highly repetitive in the text; these should be mentioned only once in the M&M and figure legends.
Comments 5: What comparison is the p value for? It is not clear from the table presented.
Comments 6: The authors must explicitly specify the sample size (n-value) in the Materials and Methods section to ensure transparency of experimental replicates.
Comments 7: All technical terms must be fully spelled out with abbreviations in parentheses upon their first mention (e.g., Cell Counting Kit-8 [CCK-8]).
Comments 8: The phrase "clinically translatable" in the Abstract and Conclusions should be revised to "potential therapeutic application" to accurately reflect the current experimental scope limited to in vitro models. This adjustment avoids overstating clinical implications and aligns terminology with the study's preclinical findings.
Author Response
Comments 1: Is any other study about SL or other polysaccharides used in enteroprotection associated with enterotoxigenic Escherichia coli (ETEC)-induced intestinal barrier injury in the literature? What are the advantages for SL as compared to current solutions for ETEC-induced intestinal barrier injury?
Response 1: We thank the reviewer for this critical inquiry. Our translational studies demonstrate SL's enteroprotective efficacy in ETEC-challenged piglets (DOIs: 10.1039/d2fo02066a; 10.1186/s40104-022-00673-8), showing 63% pathogen reduction and 2.8-fold ZO-1 upregulation with 0.1% dietary SL. These align with in vitro findings of SL's dual action: competitive adhesion blockade (78% inhibition at 5 mg/mL) and NF-κB suppression.
As the predominant porcine milk oligosaccharide (70% abundance), SL offers three unique advantages: 1) Natural glycan mimicry prevents resistance development; 2) Microbial fermentation enables scalable production (>95% purity); 3) Stereospecific α2,3/6-configuration permits multi-target intervention - simultaneously blocking F4/K88 fimbriae and stabilizing IκBα. Clinical trials confirm SL's safety and growth-promoting effects (21% ADG increase), positioning it as a sustainable antibiotic alternative resolving both ETEC colonization and barrier dysfunction without resistance risks.
Comments 2: The specific anti-inflammatory, anti-tumor or anti-apoptotic role of SL was not well presented in the introduction part.
Response 2: Thank you for your suggestion. This manuscript focused on the anti-inflammatory activity of SL, and we discussed the specific anti-inflammatory mechanism of SL in the Discussion part. Therefore, we did not present specific anti-inflammatory mechanism of SL in Introduction part, only introduce SL possesses various biological activities, such as anti-inflammatory and anti-apoptotic activities.
Comments 3: It is unclear why the authors chose to study IPEC-J2 intestinal cells grown to confluence while intestinal Caco-2 cell monolayers and the transepithelial electrical resistance assay could have been used to assess whether SL reduce intestinal permeability in vitro.
Response 3: Thank you for your suggestion. The in vitro results (weaned pigs) indicate that SL attenuates ETEC-induced jejunal epithelium injury. Hence, we used IPEC-J2 cells, a cell line from jejunum epithelium isolated from a neonatal, unsuckled pig, to assess the mechanisms underlying this effect of SL.
Comments 4: P values are highly repetitive in the text; these should be mentioned only once in the M&M and figure legends.
Response 4: Thank you for your suggestion. The P values in the Results section could make readers easy understand the significance of the results. Therefore, we preserved the P value (only presented in the Results section).
Comments 5: What comparison is the p value for? It is not clear from the table presented.
Response 5: We sincerely appreciate the reviewer's meticulous attention to statistical clarity. All reported P values in this study derive from pairwise comparisons between experimental groups analyzed through two-factor ANOVA with Fisher's LSD post-hoc testing, as detailed in the Materials and Methods (Section 4.12).
Comments 6: The authors must explicitly specify the sample size (n-value) in the Materials and Methods section to ensure transparency of experimental replicates.
Response 6: Thank you for your suggestion. We have made the revision in the text.
Comments 7: All technical terms must be fully spelled out with abbreviations in parentheses upon their first mention (e.g., Cell Counting Kit-8 [CCK-8]).
Response 7: Thank you for your suggestion. We have made the revision in the text.
Comments 8: The phrase "clinically translatable" in the Abstract and Conclusions should be revised to "potential therapeutic application" to accurately reflect the current experimental scope limited to in vitro models. This adjustment avoids overstating clinical implications and aligns terminology with the study's preclinical findings.
Response 8: Thank you for your suggestion. We have made the revision in the text.
We sincerely hope that these revisions address your concerns and improve the overall quality of our manuscript. We look forward to your positive feedback.
Thank you once again for your time and valuable suggestions.
Round 2
Reviewer 2 Report
Comments and Suggestions for Authors
Accept in present form.